# Research on Intelligent Crack Detection in a Deep-Cut Canal Slope in the Chinese South–North Water Transfer Project

**Qingfeng Hu** [1], **Peng Wang** [1], **Shiming Li** [1], **Wenkai Liu** [1,*], **Yifan Li** [1], **Weiqiang Lu** [1], **Yingchao Kou** [1], **Fupeng Wei** [2], **Peipei He** [1] and **Anzhu Yu** [3]

1   College of Surveying and Geo-Informatics, North China University of Water Resources and Electric Power, Zhengzhou 450046, China
2   College of Information Engineering, North China University of Water Resources and Electric Power, Zhengzhou 450046, China
3   School of Surveying and Mapping, PLA Strategic Support Force Information Engineering University, Zhengzhou 450001, China
*   Correspondence: liuwenkai@ncwu.edu.cn; Tel.: +86-151-3710-2399

**Abstract:** The Chinese South–North Water Transfer Project is an important project to improve the freshwater supply environment in the Chinese interior and greatly alleviates the water shortage in the Chinese North China Plain; its sustainable, healthy, and safe operation guarantees ecological protection and economic development. However, due to the special expansive soil and deep excavation structure, the first section of the South–North Water Transfer Project canal faces serious disease risk directly manifested by cracks in the slope of the canal. Currently, relying on manual inspection not only consumes a lot of human resources but also unnecessarily repeats and misses many inspection areas. In this paper, a monitoring method combining depth learning and Uncrewed Aerial Vehicle (UAV) high-definition remote sensing is proposed, which can detect the cracks of the channel slope in time and accurately and can be used for long-term health inspection of the South–North Water Transfer Project. The main contributions are as follows: (1) aiming at the need to identify small cracks in reinforced channels, a ground-imitating UAV that can obtain super-clear resolution remote-sensing images is introduced to identify small cracks on a complex slope background; (2) to identify fine cracks in massive images, a channel crack image dataset is constructed, and deep-learning methods are introduced for the intelligent batch identification of massive image data; (3) to provide the geolocation of crack-extraction results, a fast field positioning method for non-modeled data combined with navigation information is investigated. The experimental results show that the method can achieve a 92.68% recall rate and a 97.58% accuracy rate for detecting cracks in the Chinese South–North Water Transfer Project channel slopes. The maximum positioning accuracy of the method is 0.6 m, and the root mean square error is 0.21 m. It provides a new technical means for geological risk identification and health assessment of the South–North Water Transfer Central Project.

**Keywords:** central line project of south-to-north water diversion; imitation ground flight; ultra-high resolution; deep learning; intelligent detection; field location

## 1. Introduction

The Central Line Project of the South-to-North Water Diversion is a mega infrastructure project implemented to optimize the spatial and temporal allocation of water resources in China and is a key project for the future sustainable development of China [1]. Due to the wide area covered by the South–North Water Diversion Project and the changing geographical conditions, the slope of the channel is affected by many factors, such as precipitation, temperature, soil quality, and settlement, where the channel section with a complex geological foundation often faces the risk of structural instability. Surface cracks on channel slopes are a visual reflection of changes in the internal geological structure of the channel and have serious consequences if they are not detected and repaired in

time [2,3]. On the one hand, the channel cracks lead to a large amount of water seepage and water loss in the process of conveying; on the other hand, the cracks gradually develop and become larger over time with the rinsing of channel water, which may also induce channel slope instability and affect the normal operation of the channel. Since the completion of the South–North Water Diversion Project, crack detection mainly relies on regular manual inspection, which also produces consequences. One is the high labor cost, and the others include unavoidable leakage, mis-inspection, and re-inspection. With the development of remote-sensing technology, we can collect high-resolution images of the whole channel of the South–North Water Transfer Project by remote sensing means, which provides a good data source for us to carry out comprehensive channel crack detection. Given this, this paper intends to take the deep excavation channel section of the South–North Water Diversion Central Project as the engineering background and proposes to carry out research on intelligent crack monitoring technology based on remote sensing and artificial intelligence, and the research results can provide new technical means for the intelligent detection of cracks in the South–North Water Diversion Channel.

However, the automatic identification of fine cracks using remote sensing is a challenging scientific task. First, the height difference of the deep excavation channel section of the South–North Water Transfer is large, and it is very difficult to obtain ultra-high-definition channel images using traditional fixed aerial high- and low-altitude photogrammetry. Second, the exploration of fine cracks has high requirements for remote-sensing image resolution, and if the relative flight height is high, it is difficult to obtain ultra-high-resolution UAV remote-sensing images, which, in turn, cannot meet the identification needs of fine cracks. Combined with the engineering characteristics of this study area, in this paper, we chose UAV ground-like flight photogrammetry technology to obtain ultra-high-resolution remote-sensing data, which can change the relative flight height according to the terrain so that the relative flight height is always the set height in the data-acquisition process, which ensures a consistent image resolution, where the resolution of the acquired image is $6000 \times 4000$ and the ground resolution can reach 0.47 cm/pixel. Third, the acquisition of ultra-high-resolution images often introduces a significant increase in the total amount of data, and the processing of massive data also brings great arithmetic challenges.

To solve the core problem of efficient and accurate detection of cracks in deep excavated canal sections of the South–North Water Transfer and automatic processing of massive data, in this paper, we adopt the ground-imitating flying technique of a UAV and deep-learning method to research the intelligent extraction of cracks in the deep excavation channels of the South–North Water Diversion. The main contributions are as follows:

(1) Introducing the ground-imitating flying technique of a UAV to obtain ultra-high-resolution remote-sensing image data of channel slopes; the image resolution can reach millimeter level, which can meet the identification needs of fine cracks;
(2) Using deep-learning image-processing methods and constructing a channel crack image dataset for intelligent, fast, and accurate acquisition of fracture information from massive, ultra-high-resolution remote-sensing images;
(3) Using unmodeled data for combined UAV navigation information and pixel information of cracks on the image, a pioneering method for rapidly locating crack fields is proposed.

## 2. Review

### 2.1. Analysis of the Current Status of the South–North Water Transfer Channel Safety Monitoring Study

The long routes and complex geological conditions of the South–North Water Transfer Central Project are subject to geological hazards from various uncertainties during operation. At this stage, the main methods for monitoring the safety of the South–North Water Transfer Channel are InSAR technology monitoring [4–6], instrumentation monitoring [7,8], and manual inspection. Remote-sensing technology has become the main method for infrastructure deformation monitoring at this stage because of its wide monitoring range,

fast imaging speed, and low influence of ground factors. Dong et al. used multi-track Sentinel-1 data for multi-scale InSAR analysis to detect deformation along the South–North Water Transfer Central Canal [9]. J.R. et al. used GAMIT/GLOBK scientific software and GNSS-r technology to process GPS data to obtain the 3D deformation trend of the dam [10]. Xie et al. proposed an end-to-end framework to monitor the deformation of the dam after completion based on open-source remote-sensing data [11]. Chen et al. proposed a method to monitor the internal deformation of the dam by continuously measuring the flexible pipes buried in the dam body deformation, which can simultaneously measure vertical settlement and horizontal displacement along the pipe direction, achieving more accurate dam-deformation monitoring [7]. All of the above methods monitor channel deformation and cannot obtain channel crack information, while most of the pre-existing manifestations of geological hazards are cracks, so obtaining ultra-high-resolution images to find cracks is a crucial task.

### 2.2. Image Fine Line Target Detection Methods

Line target detection of images [12] has been a popular research topic in the scientific community, and Line Segment Detector (LSD) [13,14], Line Band Descriptor (LBD) [15], and Hough [16,17] transform representative methods have been developed, all of which can extract line cracks on the graph. LSD targets the detection of local linear contours in images, which can be obtained with sub-pixel accuracy in linear time [14]. LBD makes it more robust to small changes in the line direction by extracting line segments in the scale space [15]. Hough transform first transforms the binary map into the Hough parameter space, and then the detection of polar points is used to complete the detection and segmentation of line targets [16]. However, the irregularity of the crack shape and the linear background of the large number of non-cracks contained on the channel slope limit the traditional methods in their ability to detect cracks on the slope accurately, comprehensively, and automatically, especially without a large amount of a priori knowledge and manual intervention.

With the development of deep learning, especially the introduction of Convolutional Neural Networks (CNNs) [18], the application of deep learning in target detection has become more refined [19–21]. For target-detection tasks, one-stage and two-stage algorithms face different application domains. Two-stage algorithms first generate all possible candidate regions on the image for detection targets by convolutional neural networks and then classify and perform boundary regression on each candidate region based on its features [22,23]. This algorithm has high accuracy, but its efficiency is limited in processing tasks facing huge amounts of data. In contrast, the one-stage algorithm eliminates the step of generating candidate regions and directly uses convolutional neural networks to classify and localize all targets of the whole image, which is faster and more suitable for real-time and near-real-time work [24,25]. Yang et al. applied SSD deep-learning networks embedded with receptive field modules for the automatic detection and classification of pavement cracks [26]. However, the base size and shape of the prior box in the network need to be set and adjusted repeatedly during the training process in conjunction with prior knowledge, resulting in weak generalization ability, which limits the detection requirements of different regions of the SSD. The Faster RCNN algorithm achieves higher accuracy in target-detection performance through two-stage networks. Compared with one-stage networks, the Faster RCNN algorithm has more obvious advantages for high accuracy, multi-scale and small-object problems. Chen et al. used grid-surveillance video to detect and analyze the motion of maintenance personnel based on the MASK-RCNN network, and the experimental results showed that the algorithm could accurately detect multiple people and obtain key features of the video content [27]. Sharma et al. proposed a model based on Faster RCNN, which eventually showed a great improvement in detecting the average accuracy by using significance detection, proposal generation, and bounding box regression [28]. The YOLO algorithm has better applicability, a low false-recognition rate of the background, and a strong generalization ability [29]. Zhang et al. proposed an automatic bridge surface crack detection and segmentation method, improving it based on

the YOLO algorithm, which finally made the method better than other benchmark methods in terms of model size and detection speed [24]. Liu et al. proposed an image-enhancement algorithm, based on which a variety of networks were tested for road crack detection, and the final one concluded that the YOLO v5 model has the best evaluation index and can perform the crack-detection task well [30]. In general, the deep-learning method has potential application in detecting fine cracks in the slopes of the South–North Water Transfer Project.

### 2.3. Rapid Geolocation Method for Images

At the present stage, field positioning of optical images generally requires null-three calculations and leveling calculations for the whole, and finally, a high-precision three-dimensional model can be obtained [31,32]. In the monitoring and management of channel slope cracks, it is unnecessary to spend a lot of time and material resources to obtain a high-precision 3D model, and it is more difficult to establish a 3D model for the whole South–North Water Transfer Central Line project. Based on the engineering background of the deep excavation channel section of the South–North Water Transfer Central Line, quickly locating the channel slope cracks for later maintenance is an urgent need.

In this study, we propose a method to identify and obtain universal transverse Mercator grid system (UTM) coordinates of channel slope cracks by combining deep learning and a UAV positioning navigation system without processing the original data. The problem of the low resolution of channel slope crack images is solved by introducing UAV ground-like flight photogrammetry technology to collect ultra-high definition remote-sensing images; the deep-learning target-detection algorithm is used to significantly reduce the time and workload of crack detection and realize the intelligent identification of crack information from remote-sensing images; the non-modeled data of UAV combining navigation information and the pixel information of cracks on the images are used to realize the accurate location of channel cracks. The accurate location of channel cracks was achieved by using the non-modeled data combining the UAV navigation information and pixel information of cracks on images.

## 3. Materials and Methods

### 3.1. The Crack Plane's Overall Right-Angle Coordinate-Acquisition Process

The intelligent detection and location method of channel slope cracks proposed in this research can be divided into three main parts. First, the UAV ground-imitating flying photogrammetry technology is used to obtain large data of ultra-high-resolution UAV remote-sensing images that can reflect the channel slope cracks. Second, according to the acquired image characteristics of the slope of the South–North Water Transfer Channel, a suitable channel slope crack detection model was selected by comparing the experimental method while considering the detection speed, detection accuracy, and size of the model. Finally, for the remote-sensing image crack detection results, the field location of the cracks was quickly obtained by combining the UAV navigation information and pixel information of the cracks. The overall process is shown in Figure 1.

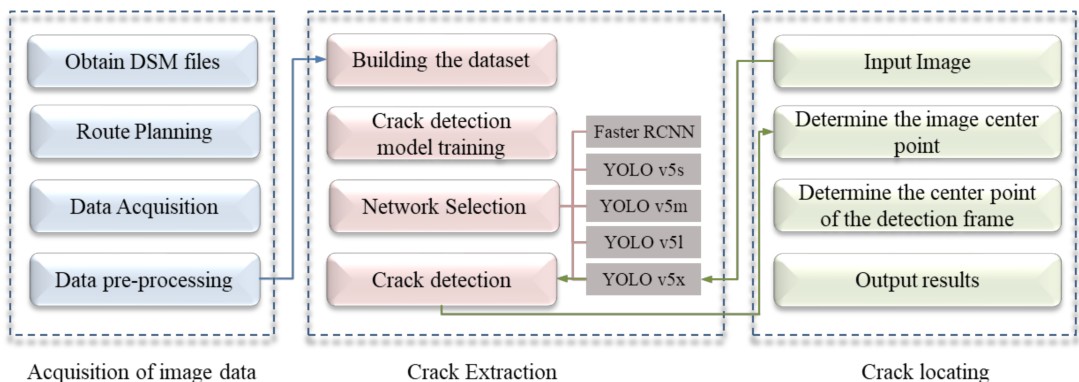

**Figure 1.** Flow chart of the method of crack detection and location.

### 3.2. UAV Ground-Imitating Flying Measurement Remote-Sensing Image Acquisition

The study area is a drain head deep-cut canal slope at the middle route of the South–North Water Transfer Central Canal, mainly located in Jiu Chong Town, Nanyang City, Henan Province. For the acquisition of ultra-high-definition image data in the study area, the first method obtained image data by flying a UAV at a fixed altitude and low altitude, with the altitude set at 30 m, but it was found that no effective fracture information could be extracted using this method. The reason for this is that due to the large topographic drop in the deep-cut canal slope, the maximum drop is as high as 47 m, and the opening width is 373.2 m. Even with a 30 m fixed flight height, the relative flight height at the bottom of the channel exceeds 70 m, and the image data obtained are unclear, causing great difficulties in fracture extraction and fracture pixel information acquisition, as shown in Figure 2a. Moreover, a fixed relative aerial height is a key prerequisite in the subsequent crack field localization process. UAV ground-mimicking technology ensures that relative altitude remains constant; thus, we introduced UAV ground-imitating flying photogrammetry technology to obtain ultra-high-resolution remote-sensing image data. The image data of the channel slope collected by the ground-imitating flying are shown in Figure 2b, and the crack information is clearly visible on the image of the channel slope.

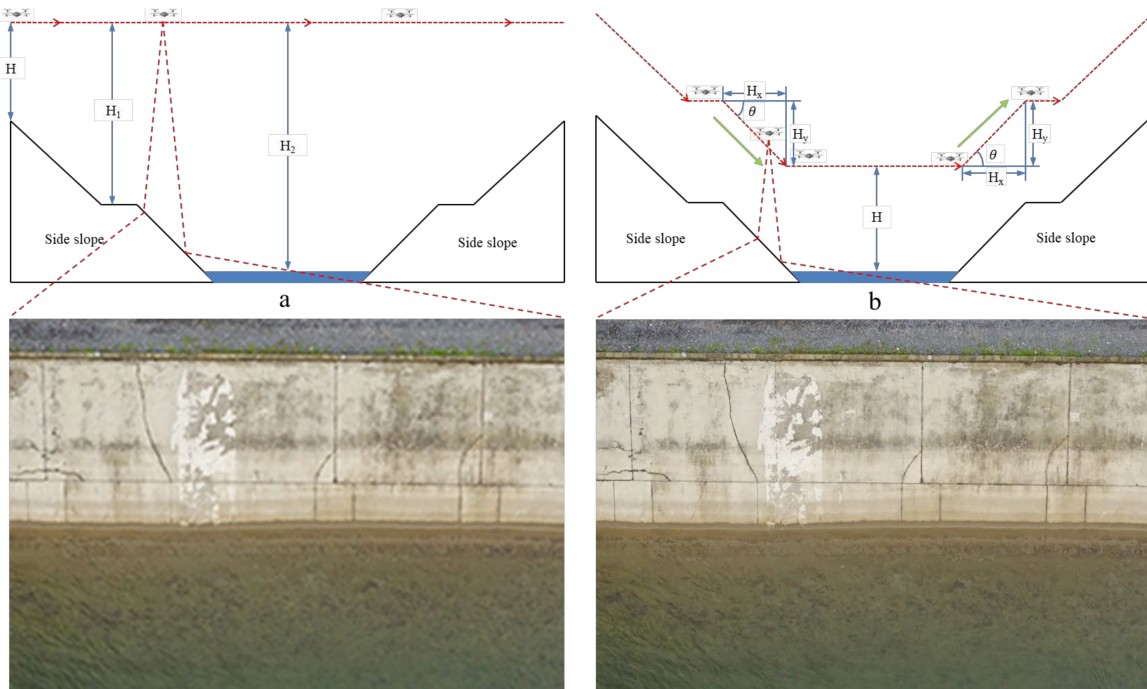

**Figure 2.** Ground-imitating and fixed high-flight acquisition images.

Ground-imitating flying, also known as terrain following, means that the UAV maintains a fixed relative height difference between the route and the 3D terrain in real time during the aerial-survey operation. This type of aerial flight makes it possible to keep the image ground resolution unaffected by terrain changes by adjusting the route with the terrain changes. The use of ground-imitating flying photogrammetry technology requires a DSM database with a certain level of accuracy before accurate ground-mimicking can be carried out. Therefore, this study uses the DJI Phantom 4RTK UAV to collect DSM terrain data in the survey area using a fixed altitude flight method, with the flight altitude set to 80 m and the heading overlap and the side overlap both set to 75%. The DSM file of the test area was obtained after data processing, as shown in Figure 3.

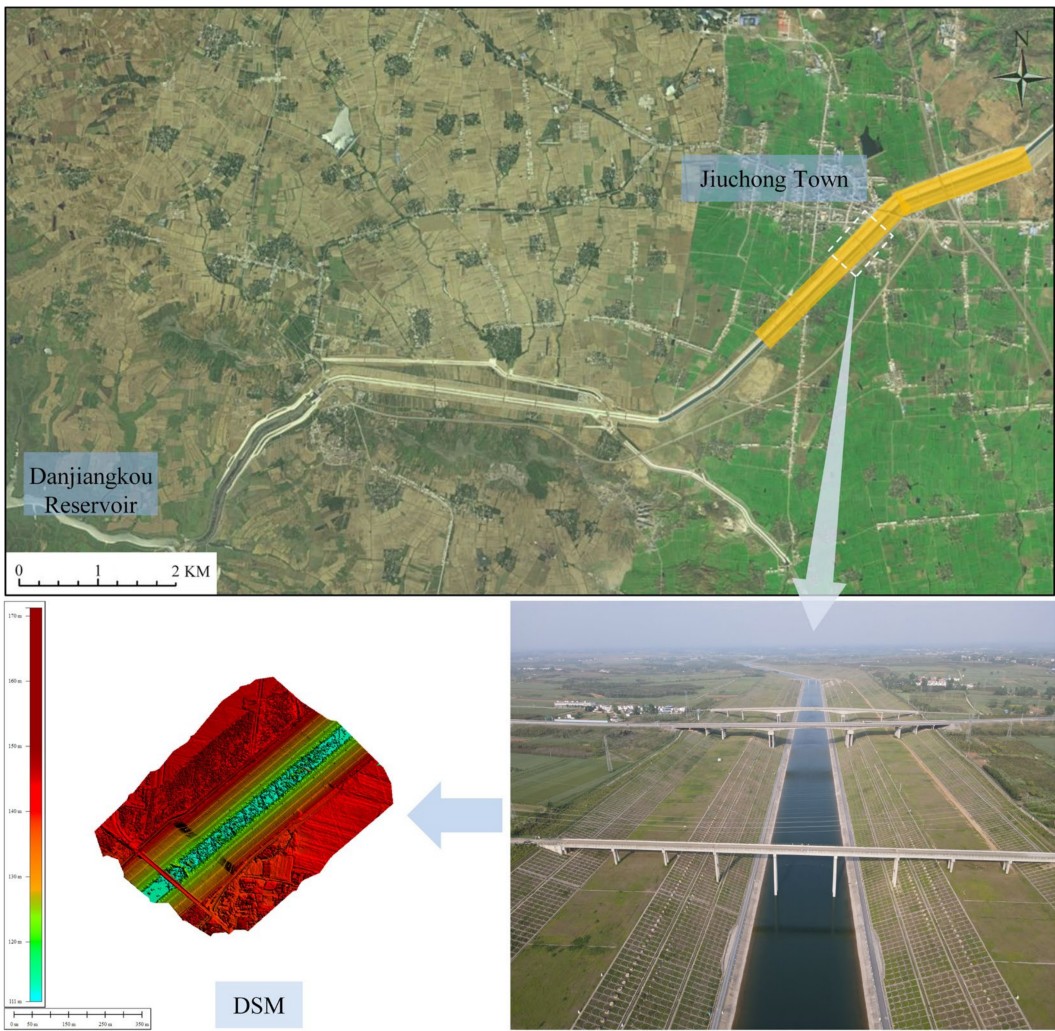

**Figure 3.** Digital surface model of the study area.

Ground-imitating flying is mainly conducted through the UAV's flight control module to control the altitude change of the UAV; when the terrain changes, the UAV generates a pitch angle θ (when θ > 0°, UAV elevation; when θ < 0°, UAV decline), causing the UAV flight altitude to change, thus, ensuring that the UAV always performs the route mission according to the fixed altitude, as shown in Figure 2b.

To ensure the accuracy of the completed 3D model, the image's control points should be evenly distributed in the survey area, and the image's control points should be distributed in a flat area without obstruction to ensure the stability and visibility of the image-control markers. There are 6 control points laid in the experimental area, and the accuracy of the control points is shown in Table 1. After the field survey, we found that the highest surface feature height in the survey area was 21 m, and there were high-voltage transmission lines on both sides of the channel slope; to ensure flight safety and the clarity of the channel slope cracks on the image, the final setting of the relative flight height of the imitation ground flight was 30 m, the heading overlap rate was 80%, and the side overlap rate was 70%. Route planning is shown in Figure 4.

Table 1. Control points' error table.

| Name | Category | Accuracy (m) | Number of Calibrated Photos | RMS of Reprojection Error (pixels) | RMS of Distances to Rays (m) | 3D Error (m) | Horizontal Error (m) | Vertical Error (m) |
|---|---|---|---|---|---|---|---|---|
| D1 | 3D | | 30 | 0.33 | 0.00552 | 0.00261 | 0.00257 | 0.00044 |
| D2 | 3D | Horizontal: | 29 | 0.13 | 0.00418 | 0.00114 | 0.00074 | 0.00086 |
| D3 | 3D | 0.01; | 26 | 0.16 | 0.00439 | 0.00185 | 0.00072 | 0.0017 |
| D4 | 3D | Vertical: | 29 | 0.2 | 0.00467 | 0.00184 | 0.00137 | −0.00123 |
| D5 | 3D | 0.010 | 32 | 0.24 | 0.0039 | 0.00296 | 0.00112 | −0.00274 |
| D6 | 3D | | 31 | 0.23 | 0.00728 | 0.0021 | 0.00172 | 0.00122 |

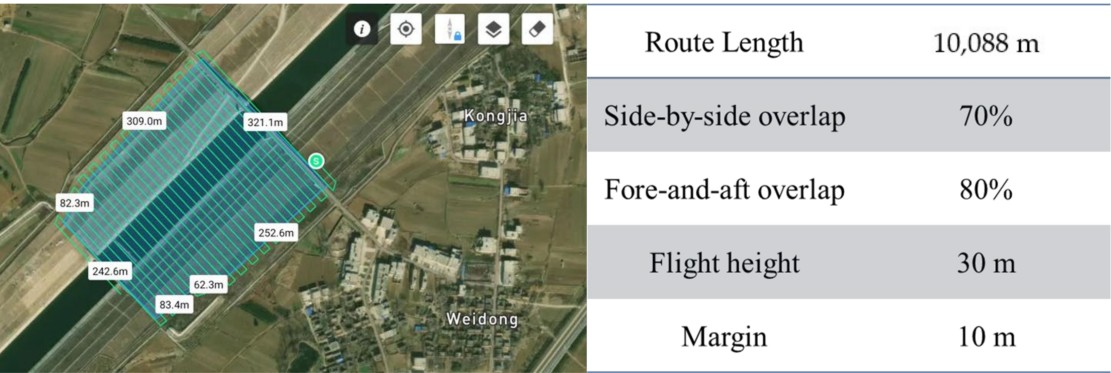

| Route Length | 10,088 m |
|---|---|
| Side-by-side overlap | 70% |
| Fore-and-aft overlap | 80% |
| Flight height | 30 m |
| Margin | 10 m |

**Figure 4.** UAV route planning and basic parameter-setting chart.

### 3.3. Channel Slope Crack-Detection Methods

YOLO v5 is one of the most stable and widely recognized target-detection methods in the current one-stage algorithm. YOLO v5 is characterized by small average-weight files, short training time, and fast inference speed. However, the YOLO v5 network is not ideal for small target detection, so we added an image-cropping module to the YOLO v5 network to crop the input images for preprocessing. Since the original image resolutions are all 6000 × 4000, the pixel coordinate values of the upper left corner point and lower right corner point of the cropped area are (2100, 1400) and (3900, 2600), respectively. Only 9% of the center area of the original image is retained, making the relative size of the cracks on the image larger and effectively solving the problem of YOLO v5's difficulty in recognizing small targets. Thus, we chose the corrected YOLO v5 network framework to conduct the remote-sensing image crack-target-detection study. Its network structure is shown in Figure 5, which consists of four main parts: input, trunk network, neck, and prediction. Mosaic data enhancement on the input side improves the detection of small targets, the addition of adaptive anchor frames improves the recall of detection, and the unification of the input image size is achieved by adaptive image scaling; the trunk network uses the CSPDarknet53 network structure to enhance the learning capability of the network and reduce the volume of the model while maintaining accuracy; the neck uses the FPN + PAN structure to achieve the goal of predicting three different scales by combining the two and aggregating parameters from different backbone layers to different detection layers; the output side uses GIOU Loss as the loss function of the Bounding box to improve the speed and accuracy of detection.

Among them, the Crop module is a cropping operation for the input image, cropping out 1800 × 1200 pixels from the center area of the original image and using the cropped image as the input data. The Focus module is a slicing operation on the input image, extracting a value from every pixel on an image so that we obtain four images, and each channel can generate four channels, and the stitched-together image becomes 12 channels from the RGB three-channel mode of the original image. Finally, we obtain a two-fold downsampling feature map with no information loss. The CBL module consists of Conv

(Convolution) + BN (Fully Connected) + Leaky_relu activation function, where the Conv module extracts the labeled features by convolutional kernel operations in the convolutional layer. The BN module readjusts the data distribution to solve the gradient problem in the propagation process. The Cross Stage Partial (CSP) module allows the model to learn more features by splitting the input into two branches and performing separate convolution operations to halve the number of channels. The SPP (Spatial Pyramid Pooling) module can convert feature maps of arbitrary size into feature vectors of fixed size, which is used to solve the problem of non-uniform size of input images.

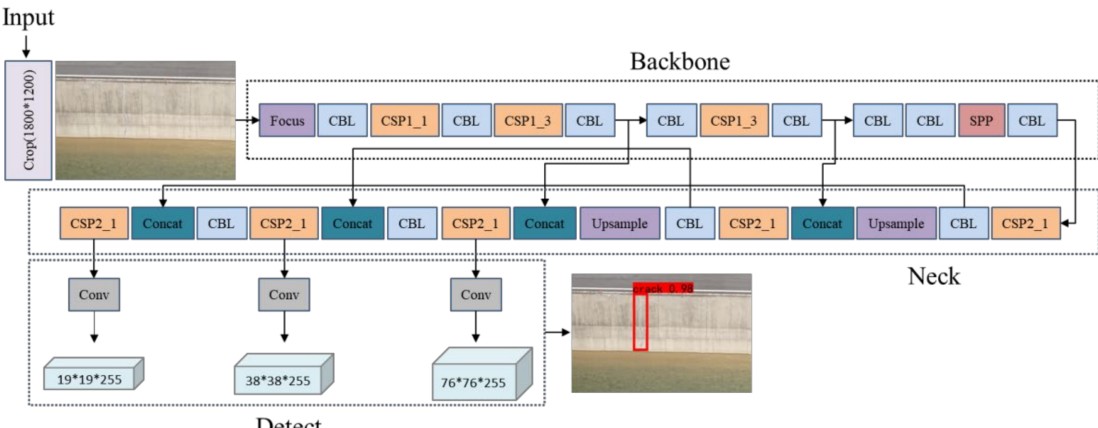

**Figure 5.** YOLO v5s network structure diagram.

YOLO v5 can be divided into four different network structures of different sizes according to different network depths and widths, YOLO v5s, YOLO v5m, YOLO v5l, and YOLO v5x; the structural parameters of the four networks are shown in Table 2. In the table, Depth_multiple and Width_multiple are the two scaling factors of the network. Depth_multiple is used to control the depth of the network, i.e., to control the number of layers of the network. Width_multiple is used to control the width of the network, i.e., to control the size of the network output channels.

**Table 2.** YOLO v5 network specifications by size.

|                | YOLO v5s | YOLO v5m | YOLO v5l | YOLO v5x |
|----------------|----------|----------|----------|----------|
| Depth_multiple | 0.33     | 0.67     | 1.00     | 1.33     |
| Width_multiple | 0.50     | 0.75     | 1.00     | 1.25     |

Another important reason for choosing the YOLO v5 network structure in this study is that YOLO v5 uses Complete Intersection over Union (CIoU) Loss to calculate the rectangular frame loss, which calculates the overlap area, centroid distance, and aspect ratio simultaneously, further improving the training efficiency, overcoming the defect that the traditional Intersection over Union (IoU) Loss can only represent the ratio of the intersection and merge of the prediction box and the real box, and cannot reflect the relative position of the prediction frame and the real frame. The equation for calculating CIoU Loss is shown in Equation (1). D denotes the distance between the prediction frame and the center point of the target frame, C denotes the diagonal length of the smallest enclosing rectangle, and $\frac{w_1}{h_1}$ and $\frac{w_2}{h_2}$ denote the aspect ratio of the prediction frame and the target frame, respectively.

$$\begin{cases} \mathrm{CIoU} = \mathrm{IoU} - \frac{D^2}{C^2} - \alpha v \\ v = \frac{4}{\pi^2}\left(\arctan\frac{w_1}{h_1} - \arctan\frac{w_2}{h_2}\right)^2 \\ \alpha = \frac{v}{1-\mathrm{IoU}+v} \end{cases} \tag{1}$$

Faster RCNN is one of the most representative algorithms in the two-stage algorithm, and the whole network can be divided into four main modules. The first is the conv layers module, which extracts the feature maps of the input image by a set of conv + relu + pooling layers. The second is the Region Proposal Network (RPN) module, which generates candidate frames. The first stage is to dichotomize the anchors, determine whether all the pre-defined anchors are positive or negative, generate the coordinate values of the four corner points of the pre-selected frames, and correct the anchors to obtain more accurate proposals. The third is the RoI Pooling module, which takes the outputs of the first two modules as input and combines the two modules to obtain a fixed-size region-feature map and output to the fully connected network for classification later. The fourth is classification and egression; this layer classifies the image by softmax and corrects the exact position of the object using edge regression. The output is the class to which the object belongs in the region of interest and the exact position of the object in the image.

To evaluate the performance of the channel slope crack-detection models selected in this study, the performance of each network model was compared and analyzed using precision (P), recall (R), the weighted average of precision and recall (F1-score, F1), and frames per second (FPS) of detected images. In general, the higher the F1 score, the more stable the model is, the higher the detection accuracy, and the higher the robustness. The algorithm for the F1 fraction is shown in Equation (2). The positive label and positive classification is denoted as true positive (TP); negative labeling and negative classification is denoted as false positive (FP); positive label but negative classification is denoted as false negative (FN); negative labels and positive classifications is denoted as true negative (TN).

$$
\begin{cases}
P = \frac{TP}{TP+FP} \\
R = \frac{TP}{TP+FN} \\
F1 = 2 * \frac{P*R}{P+R}
\end{cases}
\tag{2}
$$

*3.4. Channel Slope Crack Plane Right-Angle Coordinate-Acquisition Method*

When UAV ground-imitating flight-image-data acquisition is performed, a set of POS data will be generated simultaneously for each photo, and the POS data of some images are shown in Table 3. POS data can accurately record the 3D coordinates of the UAV and the flight altitude of the UAV when collecting aerial images. The standard POS data include the photo number; the latitude ($B_0$) and longitude ($L_0$) of the aerial point; the UAV altitude ($i$); the roll angle ($\beta$), pitch angle ($\alpha$), and heading angle (K) during flight.

**Table 3.** Example of M300 ground-like flight POS data.

| No. | $B_0$ | $L_0$ | $i$ | $i - i_0$ | $\beta$ | $\alpha$ | K |
|-----|-------|-------|-----|-----------|---------|----------|---|
| 1 | 32.6857671 | 111.7858132 | 171.845 | 30 | 1.285 | −6.077 | 47.738 |
| 2 | 32.6857792 | 111.7858263 | 171.860 | 30 | −6.346 | −7.725 | 47.407 |
| 3 | 32.6857907 | 111.7858397 | 171.804 | 30 | −5.057 | −6.735 | 48.072 |
| 4 | 32.6858062 | 111.7858586 | 171.734 | 30 | −5.725 | −7.843 | 47.720 |
| 5 | 32.6858271 | 111.7858844 | 171.596 | 30 | −3.727 | −7.573 | 47.737 |

The aerial images captured by the UAV contain information on the dimensions of the image, including the image width ($W_0$), the image height ($H_0$), and the focal length (f) of the aerial camera. Because of the influence of the pitch and roll angle, the image is also not absolutely horizontal, and the same is true for the mapped area of the image changes. Figure 6a represents the change of its mapping area when the image undergoes side deviation; i is the flight altitude and $i_0$ is the ground elevation. Since there are geometric distortions in the image and the deviation is greater in the area farther away from the image center, replacing the prediction frame with its shape center point can improve the positioning accuracy. The geometric relationship between the remote-sensing image and its mapping area is used to determine the rectangular plane coordinates of the crack point on

the image. First, the latitude and longitude $(B_0, L_0)$ of the aerial point collected in POS data are converted into UTM coordinates $(X_0, Y_0)$, and the orthographic projection of the aerial point on the image deviates from the center point of the image because the orthophoto is not horizontal. Thus, the image center point coordinates need to be corrected. As shown in Figure 6b, where point A is the vertical projection point of the aerial point on the image, point B (X, Y) is the actual center point of the image.

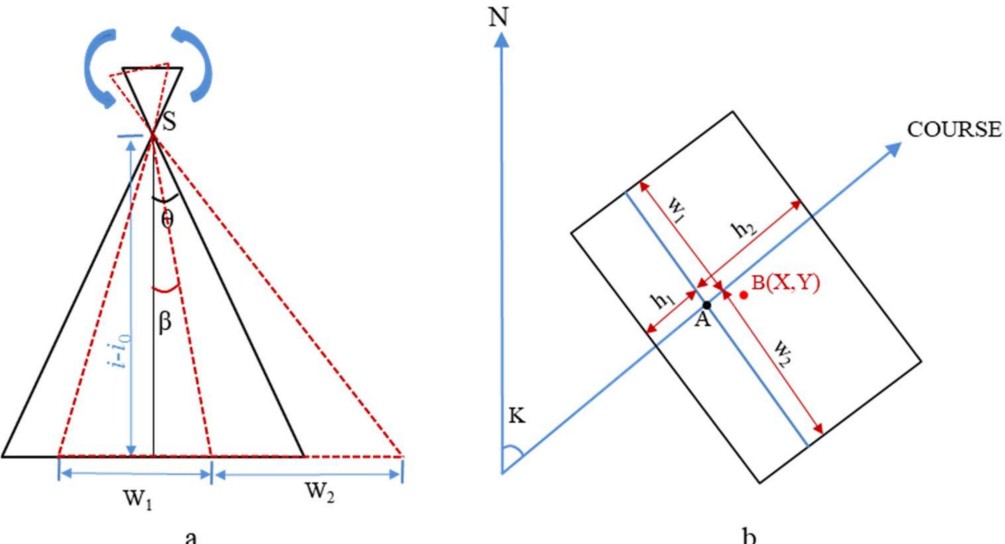

**Figure 6.** Image mapping area and image centroid correction map.

The pitch and side offset correction of the X coordinate of the image aerial point is shown in Equation (3):

$$\begin{cases} X_1 = (i - i_0) * \tan(\beta) * \sin K \\ X_2 = (i - i_0) * \tan(\alpha) * \cos K \end{cases} \tag{3}$$

The pitch and side deflection correction for the Y-coordinate of the image aerial point is shown in Equation (4):

$$\begin{cases} Y_1 = (i - i_0) * \tan(\beta) * \cos K \\ Y_2 = (i - i_0) * \tan(\alpha) * \sin K \end{cases} \tag{4}$$

Then, the actual center point coordinates of the image can be derived as shown in Equation (5):

$$\begin{cases} X = X_0 + X_1 + X_2 \\ Y = Y_0 + Y_1 + Y_2 \end{cases} \tag{5}$$

From this, the center point (X, Y) of the actual image can be determined, and then the distance and relative position of the center point of the crack on the image can be determined from the pixel value of the center point of the crack-detection frame and the pixel value of the center point of the image, and the azimuth angle from the center point to the crack point can be determined from the heading angle. According to the geometric relationship of the central projection, the field coordinates of the crack point on the ground can be determined, as shown in Figure 7.

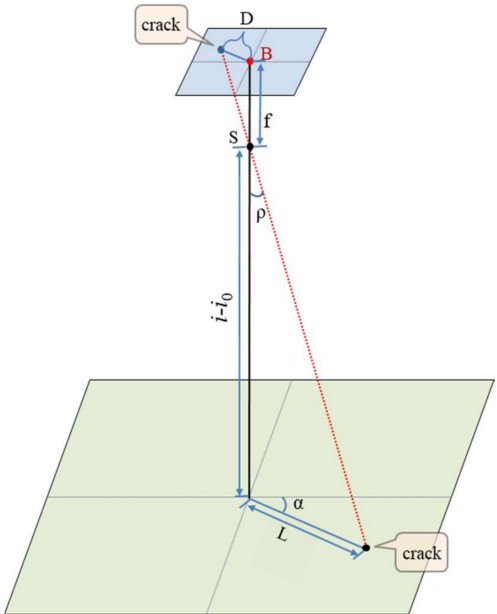

**Figure 7.** Positioning geometry schematic.

If we set the pixel value of the center point of the detection frame as (a, b), then we have:

The azimuth angle $\alpha$ from the image center to the center of the detection frame, the distance D from the image center to the center of the detection frame shape, and the distance L from the image center to the center of the crack shape on the field are obtained from Equation (6).

$$
\begin{cases}
\alpha = K + \arctan \frac{a-600}{b-900} \\
D = \sqrt{\left[\frac{(a-600)*15.6}{1200}\right]^2 + \left[\frac{(b-900)*23.5}{1800}\right]^2} \, (\text{mm}) \\
L = \frac{D*(i-i_0)}{1000*f} \, (\text{m})
\end{cases}
\tag{6}
$$

From the distance and azimuth of the center point of the image on the field to the center of the crack shape, the plane's right-angle coordinates (x, y) of the center of the crack shape on the field can be obtained from Equation (7):

$$
\begin{cases}
x = X + \Delta_X = X + D\cos\alpha \\
y = Y + \Delta_Y = Y + D\sin\alpha
\end{cases}
\tag{7}
$$

The main factors affecting positioning accuracy are, first, the existence of lateral deviation and pitch of the orthophoto causes the mapping range of the orthophoto to change. Second, the deep excavation canal section has large variations in ground undulation, resulting in inconsistent photographic scales in different areas of the image. Third, the center projection image features are geometrically distorted, and the further away from the projection center, the greater the distortion. To improve the positioning accuracy, we mainly proceed by reducing the flight height of the UAV and controlling the size of the angle between the projected ray and the plumb line; as shown in Figure 8, the lower the altitude, the smaller the angle, and the smaller the point error. Influenced by the terrain and features of the measurement area, it was finally decided that the UAV would fly at an altitude of 30 m and only retain the effective detection area in the middle of the image, accounting for 9% of the overall image. This means that the area of repeated detection is reduced, and the work process is accelerated. It can also largely attenuate the influence of geometric image distortion and effectively improve the accuracy of fracture georeferencing.

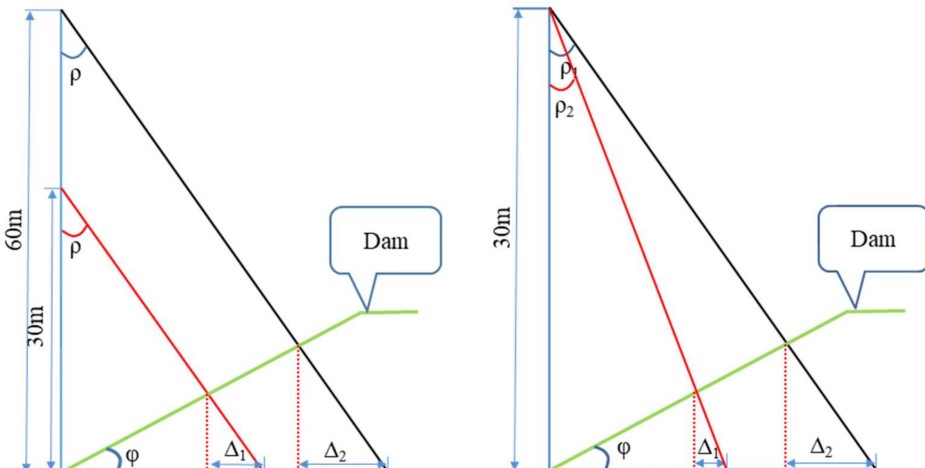

**Figure 8.** Variation of point error for different flight altitudes (**left**) and camera angles (**right**).

By comparing with the corresponding crack location information on the 3D modeling, the point accuracy of the crack information detected by using the combined navigation and positioning information of deep learning and UAV is evaluated. The 3D model of the test area is highly accurate, with an average ground resolution of 0.81cm/pixel. The position information on the model can be used as an evaluation criterion, and the detection position accuracy can be evaluated by calculating the point error $\Delta$ of both. The point error is calculated as shown in Equation (8). $\Delta_x$ is the difference between the x-coordinate of the crack obtained from the detection and the x-coordinate of the crack form center point acquired on the 3D model; $\Delta_y$ is the difference between the y-coordinate of the crack obtained from the detection and the y-coordinate of the crack form center point acquired on the 3D model.

$$\Delta = \sqrt{\Delta_x^2 + \Delta_y^2} \tag{8}$$

## 4. Experiment and Discussion

### 4.1. Experimental Data and Environment Configuration

The study was conducted in a deep-cut canal slope section of the South–North Water Diversion Project, and the RTK signal was normal during the data acquisition. The terrain of the deep-cut canal slope channel section is highly undulating, and the height difference can reach tens of meters. The width of the channel slope cracks is only a few millimeters, so traditional image acquisition means cannot clearly obtain the crack images. For this reason, this experiment introduced the ground-imitating flying technique of UAV for channel slope image acquisition for the first time. The parameters of the UAV and lens used for the measurement are shown in Figure 9.

The format of the dataset in this study is VOC2007. To improve the recognition rate of intelligent channel crack extraction using the deep-learning method, this study first preprocessed the collected image data, i.e., the original image was cropped and preprocessed, only 9% of the center area of the original image was retained, and the image size was changed to 1800 × 1200. The labeling software was then used to label the data, and a total of 15,478 cracks were labeled as the dataset, of which the training set accounted for 80%, the validation set accounted for 10%, and the test set accounted for 10%. During the labeling process, the labeling frame was ensured to fit the crack pattern as much as possible, and the labeling details are shown in Figure 10. Each annotated image generates information on the pixel coordinates of the cracks on the image for one object.

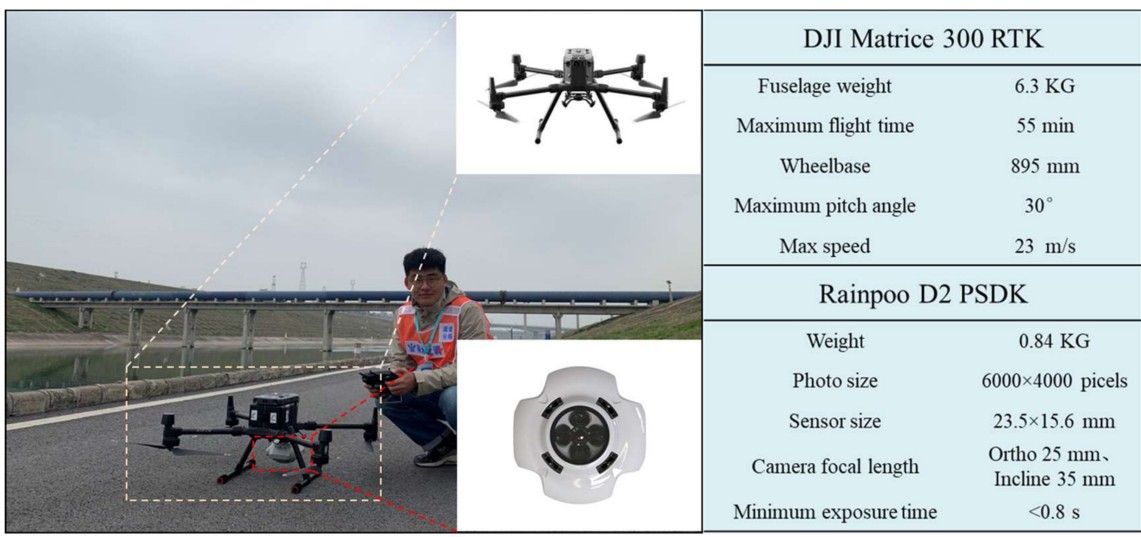

**Figure 9.** UAV survey site and main equipment specifications.

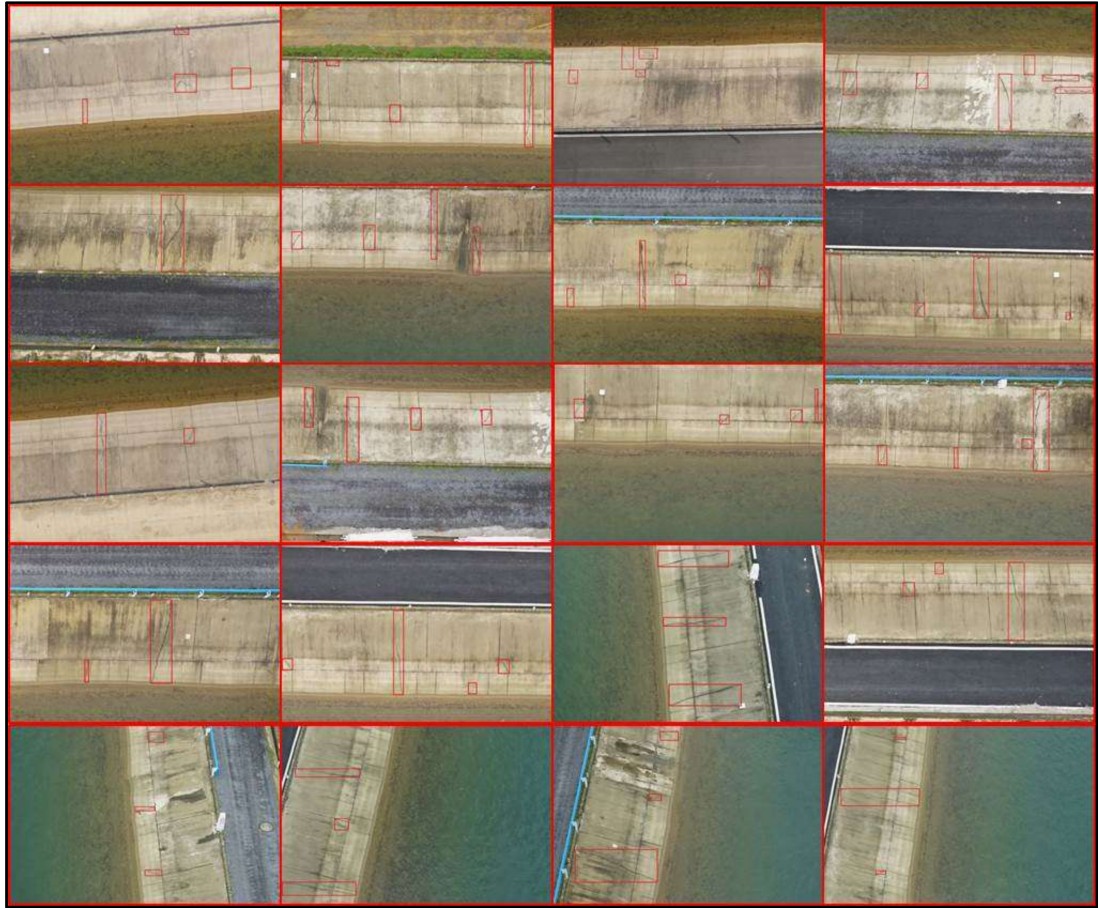

**Figure 10.** Marking details display.

In this experiment, the whole training process of the algorithm used was set to 100 epochs, and 1 epoch is the process of importing the whole dataset for complete training, with the weight decay coefficient of 0.0005 and the momentum parameter set to 0.9. Since the first fifty epoch parameters were adjusted with a large difference from the true value, the training learning rate of the first 50 epochs was set to 0.001. Each epoch was iterated

50 times, and the batch size was set to 16 to make the parameters converge faster. The learning rate of the last 50 epochs was set to 0.0001, with each epoch iterated 100 times, and the batch size was set to 8. The configuration of the experimental environment is shown in Figure 11.

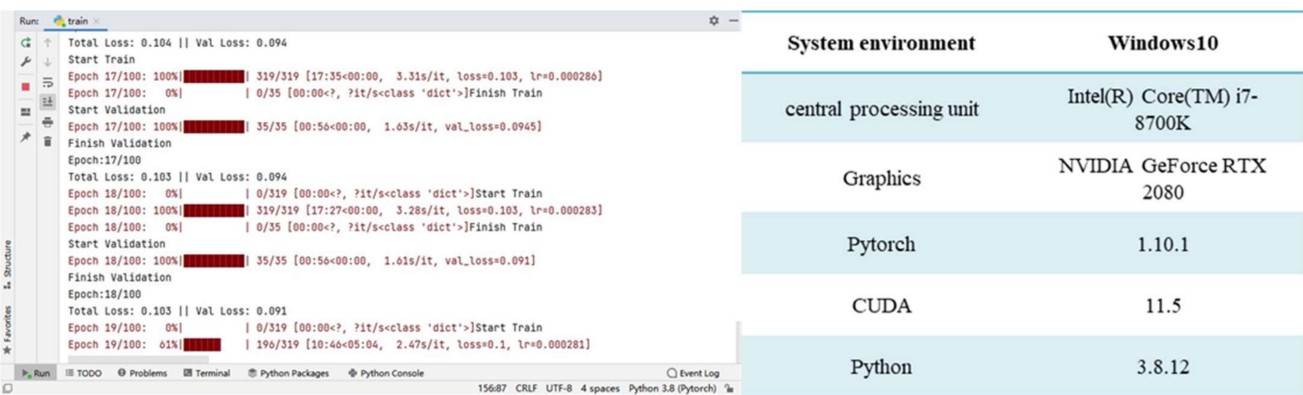

**Figure 11.** Deep-learning model training and experimental environment configuration diagram.

### 4.2. Detection Results of Slope Cracks in Different Models of Channels

To compare the crack-detection effect of each model, four models of YOLO v5 and Faster RCNN were tested under the same dataset in this study. A total of 700 preprocessed images were selected for the dataset. After the images were cropped and preprocessed, the relative size of the side slope cracks became larger, which effectively improved the small-sized crack detection accuracy of the crack-detection model. The dataset contained a total of 1945 cracks, and the performance indexes of the model were obtained, as shown in Table 4.

**Table 4.** Performance indicators for crack detection of various models.

| Models | TP | Recall | Precision | F1-Score | FPS | Model Size (M) |
|---|---|---|---|---|---|---|
| Faster RCNN | 1812 | 93.15% | 98.32% | 0.96 | 5 | 522 |
| YOLO v5s | 1626 | 83.60% | 96.85% | 0.89 | 34.5 | 27 |
| YOLO v5m | 1683 | 86.53% | 97.03% | 0.91 | 31.8 | 84 |
| YOLO v5l | 1752 | 90.08% | 96.83% | 0.93 | 28.6 | 192 |
| YOLO v5x | 1802 | 92.65% | 97.58% | 0.95 | 26.3 | 367 |

In Table 4, we can see that the Faster RCNN detection model has the best detection effect, with an F1 value of 0.96. Of the 1945 cracks included in the test, the Faster RCNN model correctly detected 1812, with a recall rate of 93.15%, which is 9.05% higher than YOLO v5s, 6.12% higher than YOLO v5m, 2.57% higher than YOLO v5l, and 0.5% higher than YOLO v5x. It is 2.57% higher than YOLO v5x and 0.5% higher than YOLO v5x; in terms of detection accuracy, Faster RCNN is somewhat higher than YOLO v5s, YOLO v5m, YOLO v5l, and YOLO v5x. Regarding detection speed, YOLO v5s, which has the smallest volume, is the fastest, with FPS up to 34.5. Regarding model volume, YOLO v5s is the smallest and has the fastest training and inference speed.

The detection results of each algorithm are shown in Figure 12. YOLO v5x and Faster RCNN have better detection results, and YOLO v5s, YOLO v5m, and YOLO v5l detection effects improved sequentially. Due to the complexity of the background of the channel slope, the detection results can be missed or wrongly detected, and the detection frame can be too small and too large, as shown in Figure 13. When cracks and expansion joints are connected, it is easy to identify them all as cracks. The water pattern paths and cracks on the channel side slopes have strong similarities in shape and tone, and the YOLO v5s and YOLO v5m algorithms do not have a strong ability to distinguish water pattern paths and

cracks. The Faster RCNN and YOLO v5x algorithms perform better in detecting channel slope cracks during the experiments, with a higher recall rate; less missed detection, wrong detection, too-small and too-large detection frames; and robustness in recognizing cracks of different shapes and sizes. The overall performance of YOLO v5x is optimal considering the accuracy of detection, speed, and model volume factors. YOLO v5x has some advantages as a channel slope crack-detection algorithm.

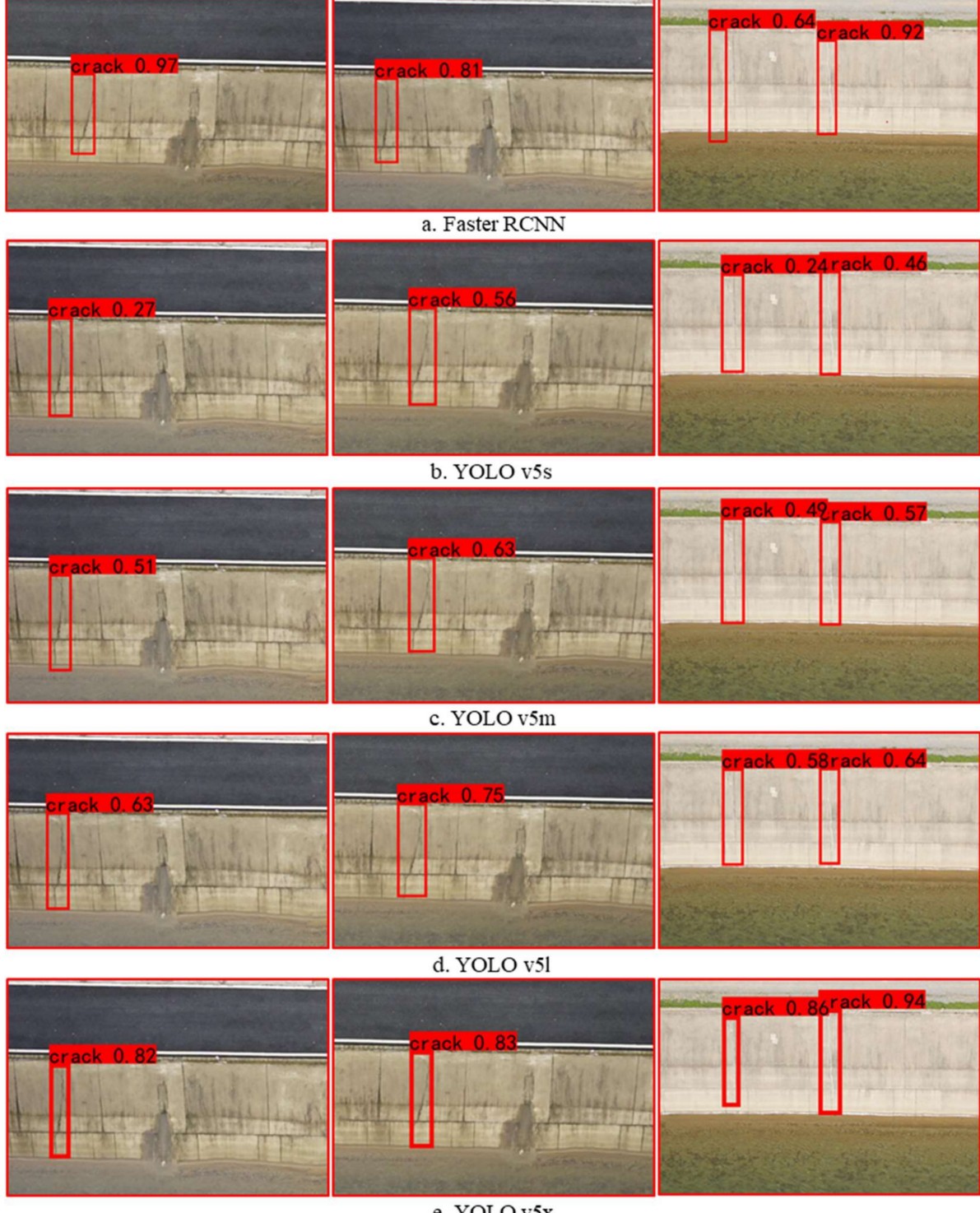

**Figure 12.** Testing results of each model.

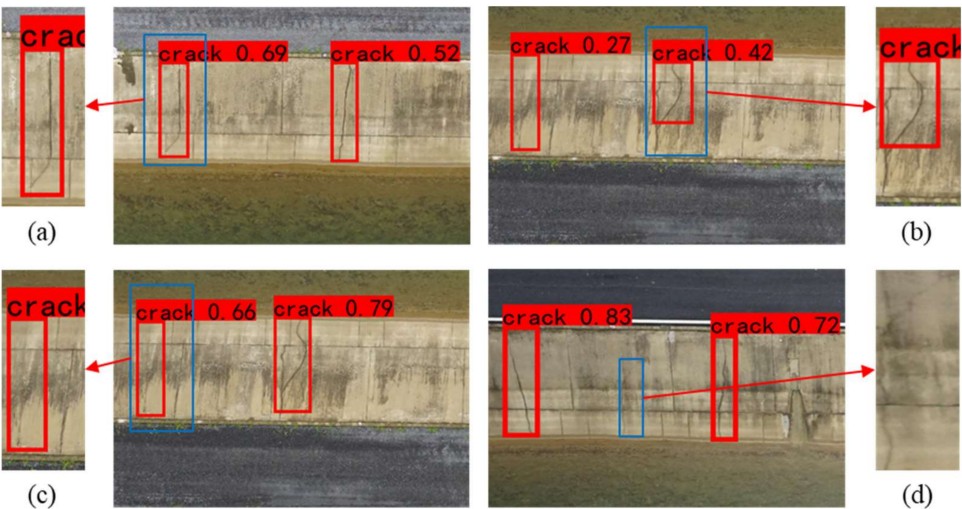

**Figure 13.** Crack-detection problem diagram. (**a**) Crack-detection frame is too large. (**b**) Crack-detection box is too small. (**c**) There is no crack in the field, and the test result shows a crack. (**d**) Cracks in the field and no cracks in the test results.

### 4.3. Application Analysis

Based on the above method, we conducted a related application study with an aerial camera sortie. To verify the feasibility of the fracture field localization method, we performed a 3D reconstruction of the whole scene and manually extracted the fractures on the model. The length of the experimental area is 730 m, the width is 270 m, and its completed 3D model is shown in Figure 14. By finding cracks on the slope of the channel of the 3D model, we determined that the area contains 74 pieces of crack information in total. By recording the crack latitude and longitude (B, L) on the 3D model and converting the latitude and longitude to UTM coordinates (X, Y), checkpoint plane accuracy was better than 0.5 pixels, the average ground resolution of the 3D model was 8.10762 mm/pixel, and the crack location information obtained through the 3D model was used for analyzing and evaluating the crack-location information obtained from the crack-detection model.

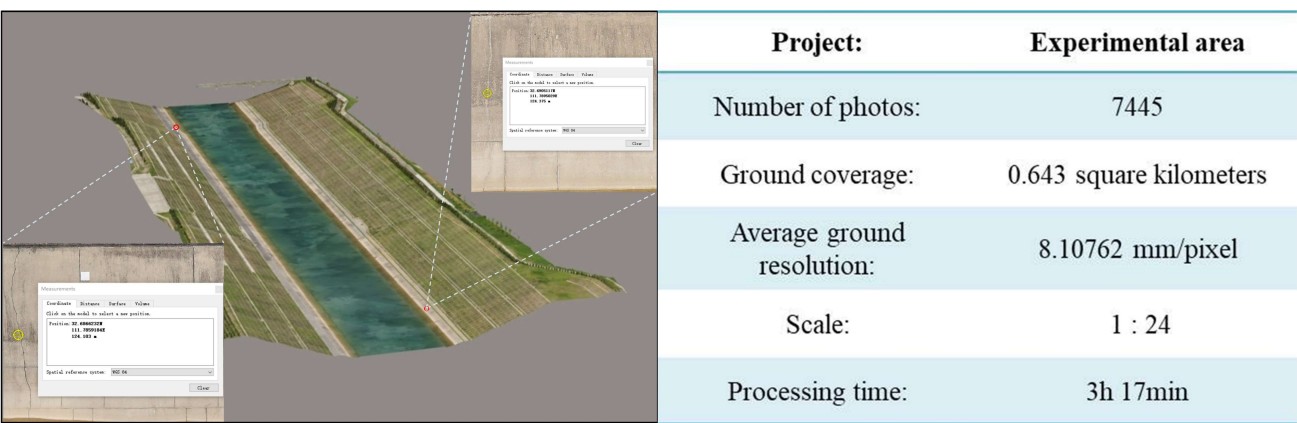

**Figure 14.** Three-dimensional model of the experimental area and its accuracy index.

The aerial image data were input into the deep-learning model for crack identification and localization, and the detection results contained information about 76 cracks, including 2 cases of detection errors, as shown in Figure 15. The field-location information of cracks on 74 3D models was recorded, the field information of crack output by deep learning was integrated, and some data are shown in Table 5. In the table, (B, L) represent the latitude and longitude of the crack center point collected from the 3D model; (X, Y) are the UTM

coordinate values transformed by the latitude and longitude of the crack points on the 3D model, respectively; (x, y) represent the rectangular plane coordinates of the corresponding cracks obtained from the model detection. The point error of each crack-detection location relative to its corresponding 3D model location can be calculated, and the value of each point error and the interval distribution of its point error are shown in Figure 16. Because the orthophoto is not horizontal and the channel slope is an inclined surface, the image has geometric distortion, which leads to errors in the field positioning of cracks. By controlling the flight height of the UAV, retaining the central area with less geometric distortion on the image as the effective detection area, and correcting the location of the image center point, we finally controlled the positioning error of the cracks to within 0.6 m, of which the point error of less than 0.3 m accounted for 73%, and the interval of the point error approximately obeyed normal distribution. Thus, it can be seen that the accuracy of the crack plane's right-angle coordinate points obtained by the model can fully meet the needs of channel slope crack maintenance, which verifies the effectiveness of this research method in the detection and positioning of cracks on the slope of the South–North Water Transfer Channel.

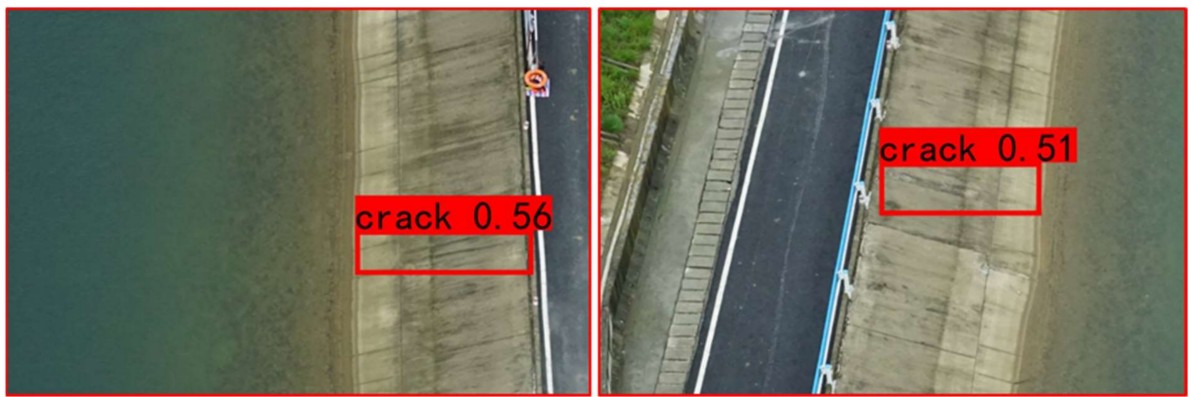

**Figure 15.** Detection of incorrect images.

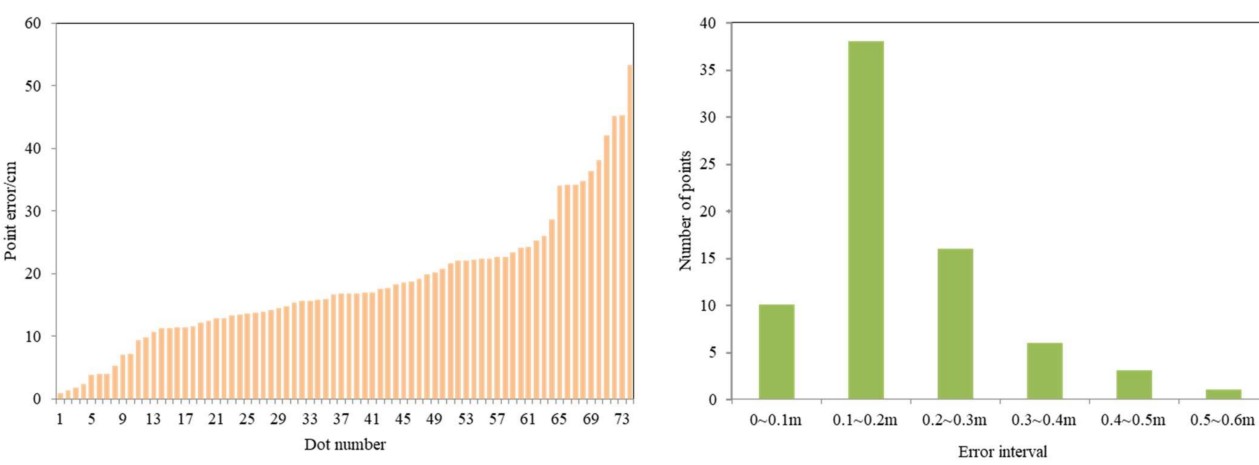

**Figure 16.** Point error interval distribution.

**Table 5.** Three-dimensional model and model's detected crack location information.

| No. | B | L | X | Y | x | y | $\Delta_x$ | $\Delta_y$ | $\Delta$ |
|-----|---|---|---|---|---|---|-----|-----|---|
| 1 | 32.68804115 | 111.7875100 | 3,648,128.981 | 228,722.352 | 3,648,128.964 | 228,722.356 | −0.017 | 0.004 | 0.017 |
| 2 | 32.68989372 | 111.7896728 | 3,648,319.436 | 228,942.128 | 3,648,319.489 | 228,942.123 | 0.053 | −0.005 | 0.053 |
| 3 | 32.68795927 | 111.7874096 | 3,648,120.599 | 228,712.183 | 3,648,120.634 | 228,712.197 | 0.035 | 0.014 | 0.038 |
| 4 | 32.69037368 | 111.7892436 | 3,648,376.040 | 228,905.875 | 3,648,376.000 | 228,905.788 | −0.040 | −0.087 | 0.096 |

**Table 5.** *Cont.*

| No. | B | L | X | Y | x | y | $\Delta_x$ | $\Delta_y$ | $\Delta$ |
|---|---|---|---|---|---|---|---|---|---|
| 5 | 32.68978919 | 111.7895845 | 3,648,308.441 | 228,932.904 | 3,648,308.545 | 228,932.956 | 0.104 | 0.052 | 0.116 |
| 6 | 32.68779782 | 111.7862056 | 3,648,111.457 | 228,597.389 | 3,648,111.567 | 228,597.410 | 0.110 | 0.021 | 0.112 |
| 7 | 32.68933022 | 111.7880129 | 3,648,268.860 | 228,780.903 | 3,648,268.704 | 228,780.837 | −0.156 | −0.066 | 0.169 |
| 8 | 32.68920729 | 111.7878633 | 3,648,256.267 | 228,765.745 | 3,648,256.111 | 228,765.777 | −0.156 | 0.032 | 0.159 |
| 9 | 32.68783291 | 111.7862416 | 3,648,115.102 | 228,601.077 | 3,648115.194 | 228,601.020 | 0.092 | −0.057 | 0.108 |
| 10 | 32.68996726 | 111.7887767 | 3,648,334.203 | 228,858.374 | 3,648,334.258 | 228,858.486 | 0.055 | 0.112 | 0.125 |
| 11 | 32.69006801 | 111.7888819 | 3,648,344.652 | 228,869.155 | 3,648,344.748 | 228,869.069 | 0.096 | −0.086 | 0.129 |
| 12 | 32.68927322 | 111.7879452 | 3,648,263.008 | 228,774.036 | 3,648,263.145 | 228,774.147 | 0.137 | 0.111 | 0.176 |

## 5. Discussion

### 5.1. Data Acquisition and Pre-Processing Methods

In this study, visible light images of channel slopes were collected by UAV. Compared with the InSAR method used in the literature ((7) and (9)) to acquire images, visible-light images can retain the hue and texture characteristics of cracks, which provides a basis for the identification and detection of cracks. In this paper, the data acquisition of the study area is carried out using the ground-imitating flying technique of a UAV. Compared with the traditional fixed-flight-height method, the ground-mimicking flight technique can ensure the consistent resolution of the images acquired in areas with different terrain, which enhances the recognition and clarity of the cracks on the complex slope background.

According to the overlap setting of the ground-imitating flying technique of a UAV, this study crops the original data for preprocessing with the same aspect ratio as the original image and retains 9% of the effective detection area in the middle part of the image. The cropped image data can cover the entire survey area to ensure that no missed detections are produced and improve the overall experiment as follows:

1. Making a dataset with cropped and preprocessed image data can improve the overall performance of the crack-detection model. The UAV flight height is set to 30 m due to the terrain characteristics of the survey area. The crack is small compared to the whole image, and the YOLO v5x network is less effective in recognizing small targets, so the features of small-sized cracks will be lost during the training process. After the data are preprocessed by cropping, the cracks are relatively larger than the cropped image, so the YOLO v5x network will be more comprehensive in learning the crack features, making the trained model more robust.

2. The accuracy of crack positioning can be improved in the crack-positioning stage. Since geometric distortion exists in all orthophotos, the geometric distortion is larger the farther away it is from the central region. In this study, the fast-localization principle based on a single image is based on the geometric relationship between the image and its mapping area, so the cropped preprocessed image retains the area with less geometric distortion, reducing the impact of geometric image distortion on the localization accuracy.

### 5.2. Processing Massive Data Using Deep Learning

Super-clear resolution images are acquired using the ground-imitating flying technique of a UAV, and the amount of data collected is huge for the South–North Water Transfer Central Line Project. For crack detection on massive image data, compared with the LSD method in the literature (13), the LBD method in the literature (15), and the Hough transform method in the literature (16), deep learning is far superior to the other methods in terms of detection speed and detection accuracy. In this paper, 15,478 cracks are labeled as the training data set, and the latest and fastest YOLO v5 algorithm and Faster RCNN algorithm are explored. YOLO v5s, YOLO v5m, YOLO v5l, YOLO v5x, and Faster RCNN are compared and analyzed in terms of their accuracy rate, recall rate, F1 score, number of frames per second of the detected images, and model volume to compare the performance

of each network model. Finally, YOLO v5x, with better overall performance, was selected as the channel slope crack-detection algorithm.

*5.3. Single-Image Positioning Method*

The detection of cracks on the slope of the channel of the South–North Water Transfer Project is different from the detection of cracks on bridges in the literature (22) and the detection of cracks on roads in the literature (26). Road crack detection and bridge crack detection can easily find the field location of the detected cracks based on the characteristic features and landforms on the images. In contrast, the long distance of the side slopes of the South–North Water Transfer Channel and the absence of obvious characteristic features make it impossible to determine the field location of cracks. At the present stage, the field location of remote-sensing images requires three empty calculations and overall leveling calculations, and finally, a high-precision, three-dimensional model is obtained. However, it is unnecessary and unrealistic to establish a 3D model for the South–North Water Transfer Central Project.

Thus, this study, based on the detection results of the YOLO v5x model, combined with the POS information corresponding to this image recorded by the UAV, first converts the latitude and longitude of aerial points in POS data to UTM coordinates, and determines the coordinates of the actual image center point (X, Y) after two corrections, replaces the location of the crack with the position of the shape center of the crack detection frame, and finds the rectangular plane coordinates corresponding to the crack by calculating the pixel value from the center point of the image to the center point of the detection frame. With this method, we can determine the plane's right angle coordinates corresponding to the crack according to the geometric relationship between the UAV image and its corresponding mapping area and improve the positioning accuracy by controlling the effective detection area in the image. The experimental results show that the point accuracy of cracks in the field is within 0.6 m, and 73% are less than 0.3 m. Thus, it is clear that the point accuracy of the plane rectangular coordinates of cracks obtained by the single-image positioning method can fully meet the needs of channel slope crack maintenance.

## 6. Conclusions

This study proposes a method combining a Positioning Navigation System with deep learning to conduct batch crack detection and positioning in the field. We can preprocess the orthophotos collected by the UAV and mark the channel side slope cracks in the preprocessed images, then, make that a training dataset. The datasets contain a total of 15,478 cracks, and by training four different networks of the YOLO v5 series and Faster RCNN, evaluating the crack field localization accuracy, the following conclusions can be drawn:

(1)  This study marks the first collection of data from the deep-cut canal slope section of the Chinese South–North Water Transfer Project by using a ground-imitating flying UAV technique, which ensures that all the images collected from the deep-cut canal slope section are of super-clear resolution and provide excellent discrimination of the channel side slope cracks. At the head of the network, the image-cropping preprocessing module is added to ensure a good detection effect for small cracks, which speeds up the overall detection rate and improves the accuracy of crack localization.

(2)  The YOLO v5x deep-learning model is selected to detect the channel slope, and the experiments show that the model outperforms other models in both detection accuracy and recall rate index. The YOLO v5x model detects cracks with a recall rate of up to 92.65%, an accuracy rate of up to 97.58%, and an F1 value of 0.95. There are fewer misses and errors in the detection process, and crack detection can be completed well.

(3)  Based on the crack-detection results from the crack-detection model, the crack-field positioning of a single image is realized by combining the image with the UAV navigation information. It is verified that the error of crack field positioning is within 0.6 m, and 73% of the crack point position error can be controlled to within 0.3 m.

The South–North Water Transfer Project is a linear feature, and the sub-meter level positioning accuracy is sufficient to provide the field position of cracks. The method of acquiring the geographical coordinates of channel side slope cracks proposed in this study can control the point position error to within 0.6 m, which is fully capable of detecting and locating the cracks of a wide range of channel slopes, reducing workloads and improving working efficiency.

This study provides new ideas and methods for the repair and inspection of cracks on the slopes of the channels of the South–North Water Transfer Project.

**Author Contributions:** Conceptualization, Q.H., P.W., S.L. and W.L. (Wenkai Liu); methodology, Q.H., P.H. and P.W.; software, P.W., Y.L., W.L. (Weiqiang Lu) and Y.K.; validation, Q.H., S.L. and F.W.; formal analysis, Q.H.; investigation, Q.H.; resources, Q.H. and A.Y.; data curation, P.W. and Y.L.; writing—original draft preparation, Q.H. and P.W.; writing—review and editing, W.L. (Wenkai Liu) and S.L.; supervision, Q.H. and W.L. (Wenkai Liu); project administration, Q.H. and W.L. (Wenkai Liu); funding acquisition, Q.H. and W.L. (Wenkai Liu). All authors have read and agreed to the published version of the manuscript.

**Funding:** This research was funded by Joint Funds of the National Natural Science Foundation of China (No. U21A20109), the National Natural Science Foundation of China (Nos. 42277478 and 52274169) and Joint Fund of Collaborative Innovation Center of Geo-Information Technology for Smart Central Plains, Henan Province and Key Laboratory of Spatiotemporal Perception and Intelligent processing, Ministry of Natural Resources (No. 212110).

**Acknowledgments:** The authors would like to thank the editor and reviewers for their contributions on the paper.

**Conflicts of Interest:** The authors declare no conflict of interest.

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
