# Peer review of "Research on Intelligent Crack Detection in a Deep-Cut Canal Slope in the Chinese South–North Water Transfer Project"

_remotesensing, doi:10.3390/rs14215384_

Round 1
Reviewer 1 Report
This study is an interesting approach to the automatic crack detection with optical images obtained with an UAV.
I strongly support your work, but I have some observations.
When you obtain the initial DSM files, you use a Phantom 4 RTK. With RTK technology, you may not need GCP's, but it is not clear, and empirically, at least 2 o 3 GCPs, are still recomended, did you think about that?
What was de resolution (cm/pix) of the images?
I suposse that you processed the images and obtained orthomosaics, what was the accuracy of the georreferenciated orthomosaics?
Visibility in figure 1 must be improved, some lines can be confusing.
Fixed and low altitude line 194
he phenomenom observed in figure 2 is called "Broken scale" (spaninsh). When you try to process some images, where the distance is different in several parts of the study area, some zones will be blurred. The methodology is interesting, why didn't you try with several flights with different heights? Maybe 3 flights in a longitudinal way?
A better explanation of the figure 6 is required
TO check de positional accuracy of the cracks, you need to georreference them, maybe GNSS. How did you get de position of the cracks? If its in the 3D model, without GCPs, you think that's enough accuracy?
The plain coordinates Where In UTM? If its, the CRS must be included in the manuscript.
Why do you use L and B for latitude and longitude?
Table 4, must include errors in X and Y components
It would be better to inlude a further statiscal analysis, the distribution of the errors, would be normal, and it doesn´t happens, why?
It would be good a more extensive literature discussión.
line 551 by calculating twice
Author Response
请参阅附件。

Reviewer 2 Report
The authors propose a work to monitor a detect cracks on the south-north water transfer project using visual information resorting to UAV acquired images dataset.
As a general comment the work shows an interesting application to a generic engineering maintenance problem, where the authors try to develop a more automated approach to solve the problem.
My only remark is that the paper does not appear to present significant scientific novelty in terms of methods and algorithms utilized. Which could be enhanced if the authors presented a more detailed benchmarked study of different methods for the detection instead of using solely YOLO versions. That could be useful for the remote sensing community, particularly if enhanced with other related work methods for infrastructure inspections.
Some minor comments.
The authors should describe much earlier in the text what do they mean by UAV ground-imitating flight.
The GSD is something that can be determined, so the authors could present a study of when the cracks are visible or not based on the GSD. I assume the authors evaluated this in flight and not to a proper theoretical study.
The related work section is poor, it can be enhanced with more related work in AI methods for infrastructure inspections.
The image cropping module is not explained, why this is done, and why that image resolution was chosen? Seems heuristically but this needs to be further detailed.
Why was the training set divided the way it did? Eighty percent usually is too much for training? Test set only 10%?
Round 2
Reviewer 2 Report
Dear author,
Despite some work that from my point of view should be in the paper. Example for me seems strange you don't have enough training data when it's something that you plan to use, enhance the SOA and so on. I recommend to accept the paper in the current format.